# UVIRT—Unsupervised Virtual Try-on Using Disentangled Clothing and Person Features

**DOI:** 10.3390/s20195647

**Published:** 2020-10-02

**Authors:** Hideki Tsunashima, Kosuke Arase, Antony Lam, Hirokatsu Kataoka

**Affiliations:** 1Computer Vision Research Team, Artificial Intelligence Research Center, National Institute of Advanced Industrial Science and Technology (AIST), Tsukuba 305-8560, Japan; hirokatsu.kataoka@aist.go.jp; 2Mercari, Inc., Tokyo 106-6188, Japan; kosuke.arase@mercari.com (K.A.); antonylam@mercari.com (A.L.)

**Keywords:** virtual try-on, image-to-image translation, unsupervised learning, GAN, disentanglement

## Abstract

Virtual Try-on is the ability to realistically superimpose clothing onto a target person. Due to its importance to the multi-billion dollar e-commerce industry, the problem has received significant attention in recent years. To date, most virtual try-on methods have been supervised approaches, namely using annotated data, such as clothes parsing semantic segmentation masks and paired images. These approaches incur a very high cost in annotation. Even existing weakly-supervised virtual try-on methods still use annotated data or pre-trained networks as auxiliary information and the costs of the annotation are still significantly high. Plus, the strategy using pre-trained networks is not appropriate in the practical scenarios due to latency. In this paper we propose Unsupervised VIRtual Try-on using disentangled representation (UVIRT). After UVIRT extracts a clothes and a person feature from a person image and a clothes image respectively, it exchanges a clothes and a person feature. Finally, UVIRT achieve virtual try-on. This is all achieved in an unsupervised manner so UVIRT has the advantage that it does not require any annotated data, pre-trained networks nor even category labels. In the experiments, we qualitatively and quantitatively compare between supervised methods and our UVIRT method on the MPV dataset (which has paired images) and on a Consumer-to-Consumer (C2C) marketplace dataset (which has unpaired images). As a result, UVIRT outperform the supervised method on the C2C marketplace dataset, and achieve comparable results on the MPV dataset, which has paired images in comparison with the conventional supervised method.

## 1. Introduction

Consumer-to-consumer (C2C) online apparel markets are a multi-billion dollar industry [1]. In C2C services, users can both buy and sell products to each other. Some of the key advantages of buying apparel through C2C services include availability of easy to find items, low price, and the large volume of merchandise. On the other hand, sellers do not need physical stores, it is easier to manage inventory, and oftentimes, they can convert unneeded goods into cash. Despite all these advantages, buyers have the problem that they cannot try on any of the apparels. Although some e-commerce (EC) websites provide services where consumers can try on apparels by delivery [2], such services are time-consuming for both the buyer and seller. Hence, virtual try-on approaches have received attention recently. Effective virtual try-on systems would enable consumers to make decisions on buying apparels more easily, improving sales.

State-of-the-art virtual try-on methods typically employ Deep Neural Networks (DNN) and 2D images rather than 3D models [3,4,5,6,7,8,9,10,11,12,13]. These virtual try-on methods also typically employ supervised learning. Supervised virtual try-on methods require ground truth data such as clothing parsing semantic segmentation masks and paired images. However, the annotation of clothing parsing semantic segmentation masks and paired images is labor intensive. Moreover, in C2C applications, supervised virtual try-on methods cannot be applied because of the lack of existing ground truth data. In C2C markets, annotating clothing parsing semantic segmentation masks and paired images is impractical since a very high volume of item listings can be added daily [14]. Therefore, virtual try-on methods which do not require annotated data are desired (e.g., unsupervised approaches). Recently, weakly-supervised virtual try-on methods have been proposed [11,12]. Although these weakly-supervised methods do not use ground truth data, they still require some annotated data or pre-trained networks which generate human dense images or parsing semantic segmentation maps. The human dense images or parsing semantic segmentation masks are used as auxiliary input information. Specifically, [12] requires clothing-parsing semantic segmentation masks to achieve virtual try-on. As for Reference [11], the method does not support clothing-only image to person translation, which is a needed functionality in virtual try-on. This is because human-parsing semantic segmentation networks require images that contain at least one person.

In this paper, we propose Unsupervised VIRtual Try-on using disentangled representation (UVIRT). UVIRT does not need any supervision, namely it does not need any annotations and does not require pre-trained networks. After the clothing features and person features are disentangled from each domain image (i.e., an only clothing image, and a person image), we can achieve unsupervised virtual try-on by manipulating the disentangled features. In Figure 1, firstly we decompose each domain image into style and content features. Next, after we switch the content features, we decode each domain image (person images vs. clothing only images) using the switched content features and each image’s original style features. In our setup, the content features contain the clothing features and the style features contain the person and background features, respectively. UVIRT can warp clothing onto a person by disentangling the clothing and the person features without annotated data or additional pre-trained networks, which would need to generate a human dense image or a human-parsing semantic segmentation map. Thus, UVIRT overcomes the problem of needing ground truth data and being difficult to apply to practical scenarios. Moreover, UVIRT significantly outperforms conventional methods in testing time because UVIRT does not use additional networks (e.g., human pose detector and human parser).

A key concept employed in our solution is *disentanglement*. Disentanglement is the idea that the inputs (e.g., images) can be broken down into factors understood intuitively by humans [15]. In particular, disentanglement has been broadly researched in unsupervised learning [16,17,18,19,20,21,22]. Inspired by this, our UVIRT approach achieves virtual try-on without ground truth data or paired images by statistically decomposing the person and the clothing features in the latent space and by being able to reconstruct each domain image from the disentangled features.

In the experiments section, we first qualitatively and quantitatively compare UVIRT with the conventional methods by evaluating on the MPV dataset [10]. We then qualitatively and quantitatively confirm that UVIRT can be applied to an in-the-wild C2C market image dataset collected by us.

We summarize the main contributions as follows:We are the first to propose a fully unsupervised virtual try-on method. Conventional virtual try-on methods use inputs such as paired images, clothing parsing semantic segmentation maps, and human dense image maps, as ground truth data or some kind of additional information. On the other hand, we do not need any additional data by employing the concept of disentanglement.We are the first to apply real-world C2C market data to virtual try-on. This is demonstrated through experiments on a real-world C2C market dataset. This is particularly challenging because the C2C market dataset has images that come from a wide variety of RGB camera types. We show that it is still possible to fuse disentangled components (person vs. clothing) features despite these challenges.Despite being completely unsupervised, our approach achieves competitive results with existing supervised approaches.Our approach is 13 times faster than the supervised method (i.e., Characteristic-Preserving Virtual Try-On Network (CP-VTON)  [4]) because ours does not use additional networks, such as the human pose detector and human parser.

The rest of the paper is organized as follows—Section 2 briefly reviews related work on virtual try-on and image-to-image translation. The proposed method is introduced in Section 3 and experimental results are presented in Section 4. Finally, conclusions are offered in Section 5.

## 2. Related Work

### 2.1. Supervised Virtual Try-on

Supervised virtual try-on methods can be mainly categorized into two general approaches, namely, GAN-based and AutoEncoders+geometrical transformation-based (hereafter, AE-based) methods.

GAN-based methods directly generate warped clothing onto people. However, GAN-based methods require paired images [13], which consist of the person wearing some target clothing and only the clothing by itself, or categorized apparel images  [6]. The annotation cost of such datasets is high.

AE-based methods [3,4,5,7,8,9,10,11,12] achieve virtual try-on through a two-stage strategy. In the first stage, the image of the target clothing is aligned with the pose of the person. In the second stage, the aligned clothing image is warped onto the person. While there have been promising results, AE-based methods need paired images and clothing parsing semantic segmentation masks. Similar to GAN-based methods, their applicability is limited in practical scenarios because of the high annotation cost. For instance, the annotation cost of semantic segmentation is generally very high in comparison with other annotations, such as class label annotation. At the same time, applying supervised virtual try-on methods to C2C market data is impractical because it is difficult for end-users to prepare annotated data with clothing parsing semantic segmentation masks. In addition, users can list any number of items on C2C markets. This means that the inventory is continually changing and has an immense variety of products. This makes application of a static training dataset for virtual try-on ineffective.

### 2.2. Image-to-Image Translation

Image-to-image translation (I2I) has been broadly researched. I2I methods are mainly categorized into supervised and unsupervised methods. Supervised methods map images based on one-to-one (e.g., pix2pix [23]), or one-to-many strategies (e.g., bicycleGAN [24]) by using paired data, such as cityscapes [25]. On the other hand, unsupervised methods map based on not only one-to-one (e.g., UNIT [26]) and one-to-many (e.g., MUNIT [27]) but also many-to-many (e.g., StarGAN [28]) strategies by using unpaired data, such as what was done in horse2zebra [29]. Although conventional unsupervised methods are prevalent due to low annotation cost, they are difficult to translate with respect to desired image regions. In our experiments, although we first employed MUNIT, it could not warp clothing to people well. Therefore, we utilize the state-of-the-art I2I assistant method “Group-wise Deep Whitening-and-Coloring Translation (GDWCT)” [30]. GDWCT can warp a clothing feature to a person as it can transfer domain-specific features well.

### 2.3. Weakly-Supervised Virtual Try-ony

To the best of our knowledge, recently, two weakly-supervised virtual try-on methods have been proposed [11,12]. We refer to them as “weakly-supervised virtual try-on methods” because these two methods still use additional data generated by pre-trained networks. The first method was proposed by Wang et al. [12]. Wang et al. achieved weakly-supervised virtual try-on by using clothing parsing semantic segmentation masks as attention for the model. While promising, annotated data was still required. The second method was proposed by Pumarola et al. [11]. They achieved weakly-supervised virtual try-on by using generated person-parsing dense images which represents 3D information and generated person parsing semantic segmentation masks. Concretely, these dense images are constructed in a UV coordinate system. In Computer Graphics (CG), the 2D orthogonal coordinate system is often used to map textures onto 3DCG models. In the 2D orthogonal coordinate system, the horizontal direction is called “U” and vertical direction is called “V”. DensePose [31] directly infers the dense image from the UV coordinate system. Concretely, the person parsing UV map is generated by a pre-trained PSPNet [32] and the person parsing semantic segmentation masks are generated by the pre-trained DensePose model [31]. However, the weakly-supervised virtual try-on method proposed by Pumarola et al. cannot be applied to model the task of applying clothing only image features to person images (dubbed clothing2person task). They can only apply their approach to transfer clothing between images that all contain people because the above two pre-trained networks only work on images containing people. We propose an unsupervised virtual try-on method (UVIRT) which achieves clothing2person generation and does not use any additional annotated data or pre-trained models. UVIRT can be broadly applied to not only aligned images of fashion e-commerce (EC) sites in general but as an added benefit, also to in-the-wild unaligned images (e.g., apparel images from C2C markets).

## 3. Methods

This section describes our proposed framework by explaining our virtual try-on strategy. We propose an unsupervised virtual try-on method (UVIRT) that does not require annotated data. Our goal is that we warp clothing onto a target person while retaining appearance features (e.g., long sleeves, characters). An overview of our model is illustrated in Figure 1. Moreover, the concrete algorithm is described in Algorithm 1.
**Algorithm 1:** Unsupervised Virtual Try-on (UVIRT)**Require:** The weight of latent $L_{1}$ loss $\lambda_{latent}$, the weight of pixel $L_{1}$ loss $\lambda_{pixel}$, the weight of whitening regularization loss $\lambda_{w}$, the weight of coloring regularization loss $\lambda_{c}$, the batchsize *m*, Adam
hyperparameters $\alpha \beta_{1},\beta_{2}$
**Require:** Initial style and content encoder parameters in domain A and B
$\phi_{s_{A}},\phi_{s_{B}},\phi_{c_{A}},\phi_{c_{B}}$, initial
decoder parameters in domain A and B$\theta_{A},\theta_{B}$, initial discriminator parameters in domain A and B
$\psi_{A},\psi_{B}$
 **while**ϕ,θ has not converged **do**
  **for**
i=1,…,m
**do**
   Sample each domain real data $x_{A}∼p_{data_{A}}\left(x\right),x_{B}∼p_{data_{B}}\left(x\right)$
   
$s_{A},s_{B}={StyleEncoder}_{\phi_{s_{A}}}\left(x_{A}\right),{StyleEncoder}_{\phi_{s_{B}}}\left(x_{B}\right)$
   
$c_{A},c_{B}={ContentEncoder}_{\phi_{c_{A}}}\left(x_{A}\right),{ContentEncoder}_{\phi_{c_{B}}}\left(x_{B}\right)$
   
$c_{\mathrm{BA}},c_{\mathrm{AB}}={GDWCT}_{\theta_{A}}\left(c_{B},s_{A}\right),{GDWCT}_{\theta_{B}}\left(c_{A},s_{B}\right)$
   
$x_{\mathrm{BA}},x_{\mathrm{AB}}={Decoder}_{\theta_{A}}\left(c_{\mathrm{BA}}\right),{Decoder}_{\theta_{B}}\left(c_{\mathrm{AB}}\right)$
   
$L_{D_{adv}}^{A},L_{G_{adv}}^{A}\leftarrow \frac{1}{2}{\left(D_{\psi_{A}}\left(x_{A}\right)-1\right)}^{2}+\frac{1}{2}{\left(D_{\psi_{A}}\left(x_{\mathrm{BA}}\right)\right)}^{2},\;\frac{1}{2}{\left(D_{\psi_{A}}\left(x_{\mathrm{BA}}\right)-1\right)}^{2}$
   
$L_{D_{adv}}^{B},L_{G_{adv}}^{B}\leftarrow \frac{1}{2}{\left(D_{\psi_{B}}\left(x_{B}\right)-1\right)}^{2}+\frac{1}{2}{\left(D_{\psi_{B}}\left(x_{\mathrm{AB}}\right)\right)}^{2},\;\frac{1}{2}{\left(D_{\psi_{B}}\left(x_{\mathrm{AB}}\right)-1\right)}^{2}$
   
$\psi \leftarrow Adam\left(\nabla_{\psi_{A},\psi_{B}}\frac{1}{m}\sum_{i=1}^{m}L_{D_{adv}}^{A}+L_{D_{adv}}^{B},\psi_{A}+\psi_{B},\alpha ,\beta_{1}\beta_{2}\right)$
   
$s_{\mathrm{BA}},s_{\mathrm{AB}}={StyleEncoder}_{\phi_{s_{A}}}\left(x_{\mathrm{BA}}\right),{StyleEncoder}_{\phi_{s_{B}}}\left(x_{\mathrm{AB}}\right)$
   
$c_{\mathrm{BA}},c_{\mathrm{AB}}={ContentEncoder}_{\phi_{c_{A}}}\left(x_{\mathrm{BA}}\right),{ContentEncoder}_{\phi_{c_{B}}}\left(x_{\mathrm{AB}}\right)$
   
$c_{\mathrm{ABA}},c_{\mathrm{BAB}}={GDWCT}_{\theta_{A}}\left(c_{\mathrm{AB}},s_{\mathrm{BA}}\right),{GDWCT}_{\theta_{B}}\left(c_{\mathrm{BA}},s_{\mathrm{AB}}\right)$
   
$x_{\mathrm{ABA}},x_{\mathrm{BAB}}={Decoder}_{\theta_{A}}\left(c_{\mathrm{ABA}}\right),{Decoder}_{\theta_{B}}\left(c_{\mathrm{BAB}}\right)$
   
$c_{\mathrm{AA}},c_{\mathrm{BB}}={GDWCT}_{\theta_{A}}\left(c_{A},s_{A}\right),{GDWCT}_{\theta_{B}}\left(c_{B},s_{B}\right)$
   
$x_{\mathrm{AA}},x_{\mathrm{BB}}={Decoder}_{\theta_{A}}\left(c_{\mathrm{AA}}\right),{Decoder}_{\theta_{B}}\left(c_{\mathrm{BB}}\right)$
   
$L_{s}^{\mathrm{BA}},L_{s}^{\mathrm{AB}}\leftarrow {∥s_{\mathrm{BA}}-s_{A}∥}_{1},{∥s_{\mathrm{AB}}-s_{B}∥}_{1}$
   
$L_{c}^{\mathrm{BA}},L_{c}^{\mathrm{AB}}\leftarrow {∥c_{\mathrm{BA}}-c_{B}∥}_{1},{∥c_{\mathrm{AB}}-c_{A}∥}_{1}$
   
$L_{cyc}^{\mathrm{ABA}},L_{cyc}^{\mathrm{BAB}}\leftarrow {∥x_{\mathrm{ABA}}-x_{A}∥}_{1},{∥x_{\mathrm{BAB}}-x_{B}∥}_{1}$
   
$L_{i}^{\mathrm{AA}},L_{i}^{\mathrm{BB}}\leftarrow {∥x_{\mathrm{AA}}-x_{A}∥}_{1},{∥x_{\mathrm{BB}}-x_{B}∥}_{1}$
   
$\Sigma_{c}^{B},\Sigma_{c}^{A}=ComputeCovariance\left(c_{B}\right),ComputeCovariance\left(c_{A}\right)$
   
$R_{w}^{B},R_{w}^{A}\leftarrow {∥\Sigma_{c}^{B}-I∥}_{1},{∥\Sigma_{c}^{A}-I∥}_{1}$
   
$UD^{A},UD^{B}=s_{A}^{CT},s_{B}^{CT}$
   
$D^{A},D^{B}=ComputeL_{2}Norm\left(s_{A}^{CT}\right),ComputeL_{2}Norm\left(s_{B}^{CT}\right)$
   
$U^{A},U^{B}=\frac{UD^{A}}{D^{A}},\frac{UD^{B}}{D^{B}}$
   
$R_{c}^{A},R_{c}^{B}\leftarrow {∥U_{A}^{T}U_{A}-I∥}_{1},{∥U_{B}^{T}U_{B}-I∥}_{1}$
   
$\phi ,\theta \leftarrow Adam(\nabla_{\phi_{s_{A}},\phi_{s_{B}},\phi_{c_{A}},\phi_{c_{B}},\theta_{A},\theta_{B}}\frac{1}{m}\sum_{i=1}^{m}\lambda_{lantent}\left(L_{s}^{\mathrm{BA}}+L_{s}^{\mathrm{AB}}+L_{c}^{\mathrm{BA}}+L_{c}^{\mathrm{AB}}\right)+$
   
$\lambda_{pixel}\left(L_{cyc}^{\mathrm{ABA}}+L_{cyc}^{\mathrm{BAB}}+L_{i}^{\mathrm{AA}}+L_{i}^{\mathrm{BB}}\right)+\lambda_{w}\left(R_{w}^{B}+R_{w}^{A}\right)+\lambda_{c}\left(R_{c}^{B}+R_{c}^{A}\right)+L_{D_{adv}}^{B}+L_{G_{adv}}^{B},$
   
$\phi_{s_{A}}+\phi_{s_{B}}+\phi_{c_{A}}+\phi_{c_{B}}+\theta_{A}+\theta_{B},\alpha ,\beta_{1}\beta_{2})$
  
**end for**
 **end while**


### 3.1. Model

Inspired by MUNIT [27], UVIRT first extracts the content (clothing) features $c_{A},c_{B}$ in a shared latent space *C* and the style features (person and background) $s_{A},s_{B}$ from the clothed person image $x_{A}$ and the clothing only $x_{B}$, respectively. Next, although we want to reflect the style features to the content features, inspired by MUNIT, the content and style features highly are entangled when using Adaptive Instance Normalization (AdaIN) [33]. Hence, we apply the style features $s_{A},s_{B}$ to the exchanged content features $c_{B},c_{A}$ by using Group-wise Deep Whitening-and-Coloring Transformation (GDWCT) [30] in order to not entangle the content and style features. Briefly, after the content feature gets whitened, using the mean of the content features (Group-wise Deep Whitening Translation: GDWT), the style feature is applied to the whitened content feature (Group-wise Deep Coloring Translation: GDCT). Finally, UVIRT can perform virtual try-on by reconstructing the person image (i.e., $x_{\mathrm{BA}}$) from the switched content features transfered by GDWCT $c_{\mathrm{BA}}$ between the clothed person image $x_{A}$ and the clothing only image $x_{B}$.

### 3.2. Loss Functions

Our model loss function is the same as in the GDWCT paper [30]. We describe the ten loss functions and two regularization terms in the following text. Note that, for the sake of simplicity, we describe things for only one domain, that is, the data flow of the transformation from domain B to domain A.

#### 3.2.1. Adversarial Loss

We introduce two adversarial losses. All adversarial losses employ the LSGAN loss [34]. We compute the LSGAN loss between an original image $x_{A}$ and a virtual try-on image $x_{\mathrm{BA}}$. These adversarial losses encourage generating high fidelity images. The adversarial loss terms are as follows,
(1)$$\begin{array}{cc} L_{D_{adv}}^{A} & =\frac{1}{2}E_{x_{A}}\left[{\left(D\left(x_{A}\right)-1\right)}^{2}\right]+\frac{1}{2}E_{x_{\mathrm{BA}}}\left[{\left(D\left(x_{\mathrm{BA}}\right)\right)}^{2}\right] \end{array}$$
(2)$$\begin{array}{cc} L_{G_{adv}}^{A} & =\frac{1}{2}E_{x_{\mathrm{BA}}}\left[{\left(D\left(x_{\mathrm{BA}}\right)-1\right)}^{2}\right], \end{array}$$
where $L_{D_{adv}}^{A}$ is the adversarial loss of the discriminator, $L_{G_{adv}}^{A}$ is the adversarial loss of the generator, D(·) is the output of discriminator.

#### 3.2.2. Reconstruction Loss

We introduce eight reconstruction losses. All reconstruction losses are based on the L1 norm. First, we compute the L1 loss between the encoded style feature $s_{A}$ and the re-encoded style feature $s_{\mathrm{BA}}$ from the transferred image $x_{\mathrm{BA}}$. It encourages the model to warp the disentangled clothing features of the reference image $s_{A}$ to the translated image $x_{\mathrm{BA}}$. The first reconstruction loss term is as follows,
(3)$$\begin{array}{c} L_{s}^{\mathrm{BA}}=E_{x_{A}}\left[{∥s_{\mathrm{BA}}-s_{A}∥}_{1}\right]. \end{array}$$

Second, we compute the L1 loss between the encoded content feature $c_{A}$ and the re-encoded content feature $c_{\mathrm{BA}}$ from the transferred image $x_{\mathrm{BA}}$. It encourages the model to maintain the disentangled person and background features $c_{A}$ after being translated $c_{\mathrm{BA}}$. In the case of domain B, to maintain the disentangled background features, the second reconstruction loss term is as follows,
(4)$$\begin{array}{c} L_{c}^{\mathrm{BA}}=E_{x_{B}}\left[{∥c_{\mathrm{BA}}-c_{A}∥}_{1}\right]. \end{array}$$

Finally, we compute the L1 loss between the original image $x_{A}$ and the cyclic reconstructed image $x_{\mathrm{ABA}}$ from the re-encoded style feature $s_{\mathrm{BA}}$ and content feature $c_{\mathrm{AB}}$. Moreover, we compute the L1 loss between the original image $x_{A}$ and the reconstructed image $x_{\mathrm{AA}}$ from the encoded style feature $s_{A}$ and content feature $c_{A}$, vice versa. This loss is the cycle consistency loss [29] which encourages image-level reconstruction. The final reconstruction loss terms are as follows,
(5)$$\begin{array}{cc} L_{cyc}^{\mathrm{ABA}} & =E_{x_{A}}\left[{∥x_{\mathrm{ABA}}-x_{A}∥}_{1}\right] \end{array}$$
(6)$$\begin{array}{cc} L_{i}^{\mathrm{AA}} & =E_{x_{A}}\left[{∥x_{\mathrm{AA}}-x_{A}∥}_{1}\right]. \end{array}$$

#### 3.2.3. Regularization

We introduce two regularization terms for GDWCT. In the GDWT phase, we encourage the covariance matrix $\Sigma_{c}^{B}$ of the content feature $c_{B}$ to be the identity matrix *I* so that the whitened content feature $c_{w}$ is encouraged to $c_{w}=c_{B}-c_{\mu }$ where $c_{\mu }$ is mean of the content feature. This procedure not only removes the need for an eigendecomposition but improves the backpropagation time. The first regularization term is as follows,
(7)$$\begin{array}{c} R_{w}=E\left[{∥\Sigma_{c}-I∥}_{1}\right]. \end{array}$$

Next, in the GDCT phase, we want to apply the covariance matrix $\Sigma_{s}^{B}$ of the style feature $s_{B}$ to the whitened content feature $c_{w}$. However, this procedure incurs expensive processing time and long backpropagation time. Therefore, the covariance matrix $\Sigma_{s}^{B}$ of style feature $s_{B}$ is decomposed to $UDU^{T}$. The colored style feature $s^{CT}$ processed by multi layer perceptron is decomposed to UD, where the *i*-th column vector $u_{i}$ of *U* is the unit vector and *D* is the diagonal matrix, that is, $s^{CT}=UD$. *D* corresponds to the $L_{2}$ norm of each column vector of $s^{CT}$. Next, UD is divided by *D*. Consequently, we can acquire *U*. *U* of the colored style feature has to be encouraged to be the orthogonal matrix. The second regularization term is as follows,
(8)$$\begin{array}{c} R_{c}=E_{s}\left[{∥U^{T}U-I∥}_{1}\right]. \end{array}$$

GDWCT employs a ResNet like decoder with multiple channels. To improve processing speed, we group these channels into batches of 32.

#### 3.2.4. Total Loss

Our model including generator (i.e., encoders and decoders) and discriminator is optimized in an end-to-end manner. The full objectives are as follows,
(9)$$\begin{array}{cc} L_{D}= & L_{D_{adv}}^{A}+L_{D_{adv}}^{B} \end{array}$$
(10)$$\begin{array}{cc} L_{G}= & L_{G_{adv}}^{A}+L_{G_{adv}}^{B}+\lambda_{\mathrm{latent}}\left(L_{s}+L_{c}\right)+\lambda_{\mathrm{pixel}}\left(L_{cyc}+L_{i}\right)+\lambda_{w}R_{w}+\lambda_{c}R_{c}, \end{array}$$
where $\lambda_{latent}$ is the coefficient of latent-level reconstruction losses and $\lambda_{pixel}$ is the coefficient of image-level reconstruction losses. We set the value of $\lambda_{\mathrm{latent}}=1$, $\lambda_{\mathrm{pixel}}=10$, $\lambda_{w}=0.001$, $\lambda_{c}\;=\;10$, respectively.

### 3.3. Collecting C2C Market Datasets

We propose a new virtual try-on dataset built from a real-world C2C marketplace, that is, from Mercari [35] (hereafter, C2C market dataset). Specifically, we collected in-the-wild clothing images consisting of 539,546 ladies top clothing 96 × 96 resolution images, organized into 3389 categories. These images come from a very wide variety of RGB camera types because the marketplace itself has millions of users. The final resolution of the C2C market images is 128 × 96 due to adding white margins to the images. For testing virtual try-on, we also couple our collected clothing only images with 10,322 front-view person images from the Multi-Pose Virtual try-on datasets [10]. In Figure 2, we can see that the C2C market dataset images are not aligned (e.g., hanged, folded). In Section 4.6, we investigate such characteristics of the dataset.

## 4. Experiments

### 4.1. Evaluation Criteria

Although conventional methods use the inception score [36] as their evaluation criterion, we do not use the inception score in our experiments. This is because the inception score measures the diversity and distinguishability of the generated images when using the Inception-v3 model. In virtual try-on evaluation, the inception score is not an appropriate evaluation criterion because it can only quantify how classifiable clothing are. Furthermore, our aim is not to randomly generate a set of images with a large class variation as is usually done in the Generative Adversarial Networks literature. Instead, we aim to generate virtual try-on images in a consistent manner. Given a person, we wish to apply a new article of clothing on the same person. Thus we choose more suitable metrics in our experiments.

#### 4.1.1. Learned Perceptual Image Patch Similarity (LPIPS)

We use the Learned Perceptual Image Patch Similarity (LPIPS) [37] to measure the perceptual similarity between any given synthesized image and its ground truth image similar to how Structured Similarity Index Measure (SSIM) [38] is used. LPIPS is very similar to human perception. However, LPIPS cannot be applied to the C2C market dataset because LPIPS needs ground truth data. LPIPS is defined as
(11)$$\begin{array}{c} d\left(x,x_{0}\right)=\sum_{l}\frac{1}{H_{l}W_{l}}\sum_{h,w}{∥w_{l}⊙\left({\hat{y}}_{hw}^{l}-{\hat{y}}_{0hw}^{l}\right)∥}_{2}^{2}, \end{array}$$
where *x* is the ground truth image, $x_{0}$ is the generated image, *l* is the number of the layers of a pre-trained network (in our case, AlexNet [39]), H,W is the height and width of the images respectively, $w_{l}$ are used in scaling the activations channel-wise and computing the $\ell_{2}$ distance and ${\hat{y}}^{l},{\hat{y}}_{0}^{l}$ the are output of the layer of the pre-trained network (AlexNet).

#### 4.1.2. Fréchet Inception Distance (FID)

We also use the Fréchet Inception Distance (FID) [40]. The generated images are high fidelity perceptually when the FID is low, because the FID has been found to be consistent with human judgement [40]. The FID measures the discrepancy between any given real data distribution and generated data distribution. The FID uses 2nd order information of the intermediate layer of the pre-trained Inception-v3 model. On its own, the Fréchet Distance [41] is the Wasserstein-2 distance between two distributions $p_{1}$ and $p_{2}$ assuming they are both multivariate Gaussian distributions. The FID is defined as
(12)$$\begin{array}{cc} F\left(p_{1},p_{2}\right)= & ∥\mu_{p_{1}}-\mu_{p_{2}}{∥}_{2}^{2}+trace\left(C_{p_{1}}+C_{p_{2}}-2{\left(C_{p_{1}}C_{p_{2}}\right)}^{\frac{1}{2}}\right), \end{array}$$
where $\mu_{p_{1}}$ and $C_{p_{1}}$ are the mean and covariance of the samples from $p_{1}$ and similar for $p_{2}$. In this paper, we compute the FID 10 times using random generated paired images because the FID of each generated image is different.

### 4.2. Implementation Details

#### 4.2.1. Conventional Dataset

We also conduct experiments to compare conventional supervised virtual try-on methods and our UVIRT method on the Multi-Pose Virtual try-on (MPV) datasets collected by Dong et al. [10]. It contains 35,687 person and 13,524 clothing images at 256 × 192 resolution. Dong et al. split them into train/test sets of size 52,236 and 10,544, respectively. We chose the front-view person and top clothing image pairs from the MPV dataset, resulting in 10,322 train and 2171 test pairs.

#### 4.2.2. Setting

In our experiments for comparing between conventional methods and UVIRT on MPV dataset, we trained UVIRT for 500k iterations. UVIRT consists of four encoders, two decoders, and two discriminators. The four encoders and the two decoders make up the generator and we trained using the Adam optimizer [42], with α=0.0001 (i.e., learning rate), $\beta_{1}=0.5$, $\beta_{2}=0.999$, and a batch size of 4. The two discriminators and the one classifier were trained using the Adam optimizer with the same hyperparameters as the generator. Note that we trained UVIRT with the paired image setting for fair comparison because CP-VTON can be only trained with the paired image setting.

In experiments using the C2C market dataset, we trained UVIRT for 500 k iterations. The architecture, optimizers, and hyperparameters were the same as the above experiments except for a batch size of 8. Note that we trained UVIRT with the unpaired setting because the C2C market dataset does not have paired images.

#### 4.2.3. Architecture

As shown in Figure 1, the two encoders which extract the content features are structured as a ResNet-like network, which consists of one convolutional layer, two downsample convolutional layers, and eight ResBlocks, sequentially. The two encoders which extract the style features have a fully convolutional network, which consists of one convolutional layer, four downsample convolutional layers, and a global average pooling layer, sequentially. The two decoders are structured as a ResNet-like network, which consists of the GDWCT module whose architecture is the same as in Reference [30]. The two discriminators have fully convolutional networks, which consist of three downsample convolutional layers and one point-wise convolutional layer, sequentially. Plus, the above discriminators are multi-scale discriminators [43]. Finally, the one classifier is a simple convolutional network, which consists of two downsample convolutional layers and one fully connected layer, sequentially. Note that in the C2C market dataset, we use 2171 test images which contain only clothing because we use the test person images of the MPV dataset as the person and background domain. We fix the number of the test images of the C2C market dataset because the number of person images of the MPV dataset is 2171.

### 4.3. Baseline

We adopt CP-VTON [4] as the baseline image-based virtual try-on method. First, CP-VTON estimates the parameters of geometrical transformations by concatenating the key points feature, a rough body segmentation mask and only the head image in the channel dimension, and using the only clothing images. In total, the concatenated feature has 23 channels and the only clothing images have 3 channels. The key points feature has 19 channels, the rough body segmentation mask has 1 channel and the head-only image has 3 channels. Second, CP-VTON generates a warped mask and a coarse rendered person image. Finally, CP-VTON concatenates the warped mask, coarse rendered person image, and warped clothing. As a result, CP-VTON generates a virtual try-on on image. CP-VTON employs supervised training using a clothing parsing semantic segmentation mask. Moreover, CP-VTON uses key points, a body rough segmentation mask and a head-only image as auxiliary information. On the contrary, our UVIRT directly generates a virtually tried on image using completely unsupervised training.

### 4.4. Comparison with Conventional Supervised Methods

#### 4.4.1. Qualitative Results

We first perform visual comparisons of our proposed UVIRT method with conventional supervised methods, namely CP-VTON [4], illustrated in Figure 3, which shows that our UVIRT method generates comparable results. Although the pose of the people in the images generated by our UVIRT method is altered, UVIRT naturally warps the clothing to the people because UVIRT can adequately disentangle person and clothing features. As a result, UVIRT provides competitive performance without any annotation cost.

#### 4.4.2. Quantitative Results

We also compare our proposed UVIRT method with CP-VTON quantitatively based on LPIPS and FID. Note that we virtually try on the test clothing images to the test person images based only on predetermined pairings since LPIPS can be applied to the comparison of generated images only when ground truth images are available. In Table 1, although CP-VTON achieves a lower mean LPIPS (i.e., better score) than our UVIRT method, UVIRT can generate high fidelity images. The reason for the higher LPIPS score is that LPIPS measures how a generated image corresponds to a ground truth image. Next, we measured the FID of the images generated by CP-VTON and our UVIRT method. We warp the test clothing to the random test persons 10 times since we do not need ground truth images when computing FID. We computed the mean and standard deviations of FID after we computed FID 10 times on the all test images, that is, 2171 images. In Table 1, UVIRT achieves a worse score than CP-VTON. The reason of the former deterioration is that the images generated by UVIRT exhibit some reconstruction deficiencies and distorted body parts. These results are natural in LPIPS and FID because CP-VTON is trained in a supervised manner, in comparison with our UVIRT trained in an unsupervised manner. Note that UVIRT is the first unsupervised virtual try-on method. In future work, we aim to improve with respect to these metrics.

### 4.5. Virtual Try-on on C2C Market Dataset

#### 4.5.1. Qualitative Results

We conduct experiments on the C2C market dataset and compare visual results of our UVIRT and CP-VTON in Figure 4. In this experiment, we use CP-VTON trained with the MPV dataset since CP-VTON cannot be trained in an unsupervised setting. In testing time, CP-VTON can be applied to novel dataset when anyone have person images, only clothing images, human-parsing semantic segmentation masks and pose features. Human-parsing semantic segmentation masks and pose features are generated by a pre-trained human-parser and pose estimator. In Figure 4, the images generated by CP-VTON are distorted and have color artifacts because CP-VTON warps geometrically transformed clothing onto the person. CP-VTON is affected by distribution shift as the MPV dataset is aligned by taking the white background and centering. On the other hand, our UVIRT can generate high fidelity person images with tried on clothing without aligned images since ours exchanges the disentangled person and clothing features in the latent space. Thus, our UVIRT is an effective method when ground truth and aligned images are unavailable. Although our UVIRT indicates competitive results with the supervised method, ours does share some of the same problems. First, in Figure 5a, not only did the color of the target clothing change but also the face of the generated image is not reconstructed well. Moreover, in Figure 5b, not only did the color of the clothing change but also the person changed. The reason for the above problems is that UVIRT cannot disentangle between the person and the clothing features well enough. UVIRT encodes input images to the style and content features. It is possible that the style and content features are not semantically orthogonal. Thus, the style and content features are entangled. Therefore, we must tackle the entanglement problem. Concretely, we will try using the unsupervised disentanglement techniques of the generative models [16,17,18,19,20,21,22,44] in future work.

#### 4.5.2. Quantitative Results

We compare the conventional supervised virtual try-on method CP-VTON and our unsupervised virtual try-on method UVIRT using the widely used metric, FID to verify the performance of image synthesis, summarized in Table 2. CP-VTON was trained with the MPV dataset because it is a supervised method and cannot be trained on the C2C market dataset. UVIRT was trained with the MPV dataset and the C2C market dataset for fair comparison. When both methods were trained on the MPV dataset, CP-VTON showed better performance on the C2C market dataset. However, when UVIRT was trained on the C2C market dataset (something that cannot be done for CP-VTON), UVIRT demonstrated significantly better performance. The strong point of UVIRT is that it can be trained with in-the-wild datasets. Our UVIRT also achieves more high fidelity image generation than CP-VTON when trained with the C2C market dataset. Consequently, we have demonstrated our UVIRT is effective qualitatively and quantitatively in an in-the-wild dataset virtual try-on task.

### 4.6. Ablation Study

In this section, we conduct an ablation study on the scale of the C2C market dataset. We confirm whether or not the scale of the dataset affects virtual try-on results in our unlabeled setting. In Table 3, we use the FID metric for comparison of the training data at different scales. We randomly extract the training data at 5%, 10%, 25% and 50% from the full training dataset. We compute FID in test images after we training UVIRT with each extracted training data. When we use the 50% scale setting of the training data, FID is best because UVIRT could capture the data distribution correctly from the increased amount of training data. However, when we use the 100% scale of the training data, the FID of UVIRT worsens compared with the 50% scale of the training data. We assume that UVIRT does not capture the data distribution well because more training epochs were needed in the 100% scale setting. Computational resources limited our ability to fully train the model on all the data. But we believe that even with limited resources, training with medium sized datasets can still give good results.

### 4.7. Testing Time Comparison

We compare the testing time between UVIRT and CP-VTON on the C2C market dataset. This comparison excludes the data loading time and counts only the testing time. In Table 4, the testing time is the mean forward propagation time on the 2171 test images. We use batchsize = 1. The testing time of CP-VTON includes the time needed to run a pre-trained human keypoints detector (OpenPose) [45,46,47,48] and the pre-trained human parser (JPPNet) [49,50]. CP-VTON needs the keypoints and the human-parsing semantic segmentation mask in its testing phase as well as training phase. In Table 4, UVIRT outperforms CP-VTON by about 13 times. UVIRT not only is a fully-unsupervised method but also has fast testing time.

## 5. Conclusions

In this paper, we proposed a novel virtual try-on system, namely Unsupervised VIRtual Try-on system (UVIRT). The annotation cost of the training data for virtual try-on is high because the conventional supervised approaches need annotated data such as clothing-parsing semantic segmentation masks and paired images. Therefore, we resolved the problem of the annotation cost through our unsupervised approach and by using a state-of-the-art image-to-image translation method. In addition, we can apply UVIRT to practical scenarios for virtual try-on in a real-world C2C market. Conventional supervised virtual try-on methods cannot be applied to the C2C market dataset because the accumulation of annotated data for the C2C market is very difficult. In the C2C market dataset, we outperformed a pre-trained conventional supervised virtual try-on system. Moreover, we not only constructed a new C2C market dataset but also indicate the usefulness of unsupervised virtual try-on by investigating how the size of the C2C market dataset affects performance. In future work, we will resolve the issues with distortion of the person and deficiencies in face reconstruction.

## Figures and Tables

**Figure 1 sensors-20-05647-f001:**
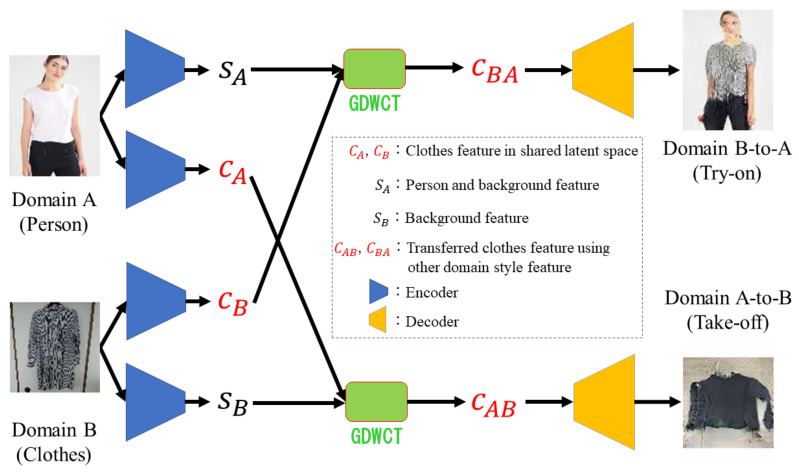
Overview of our system. Firstly, the person image and the clothing image are encoded into the content feature *c* and the style features *s*. Note that each domain’s content features $c_{A},c_{B}$ captures the clothing features, the domain A style feature $s_{A}$ captures the person and background feature and the domain B style feature $s_{B}$ captures the background feature. Next, the exchanged content features $c_{B},c_{A}$ are combined with the style features of each domain $s_{A},s_{B}$ by the Group-wise Deep Whitening-and-Coloring Transformation (GDWCT). Finally, the transformed content features $c_{\mathrm{BA}},c_{\mathrm{AB}}$ are decoded into their new domain images. Note that this figure is subset of the whole architecture. The two discriminator, $x_{\mathrm{BAB}}$, and $x_{\mathrm{ABA}}$ streams are omitted.

**Figure 2 sensors-20-05647-f002:**
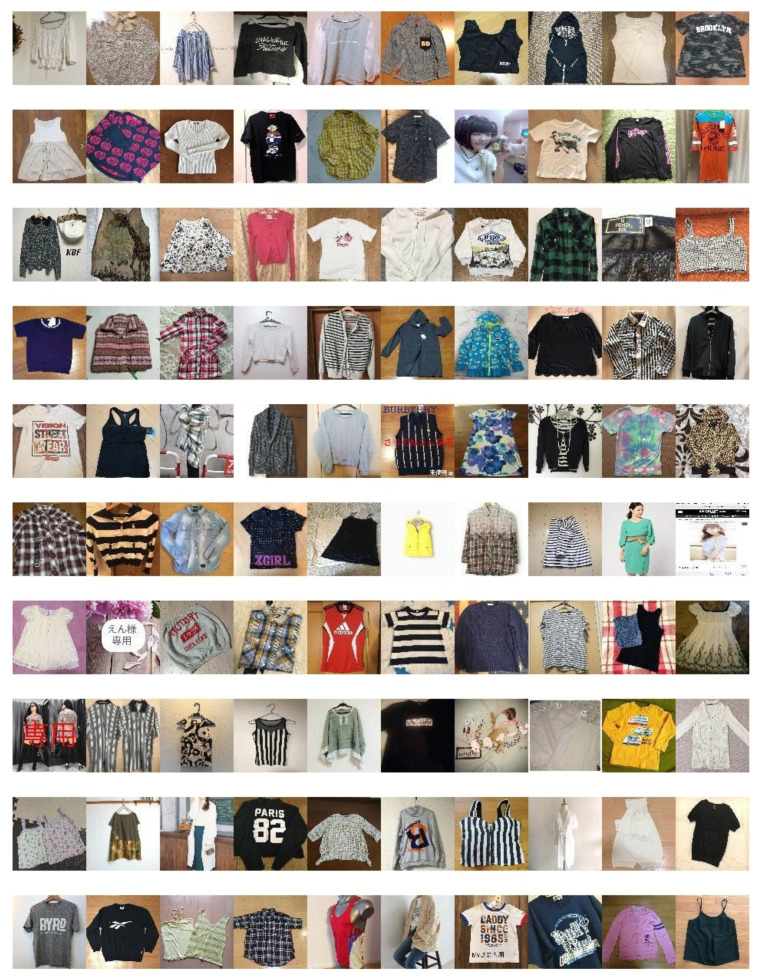
Sample images from the consumer-to-consumer (C2C) market dataset collected by us.

**Figure 3 sensors-20-05647-f003:**
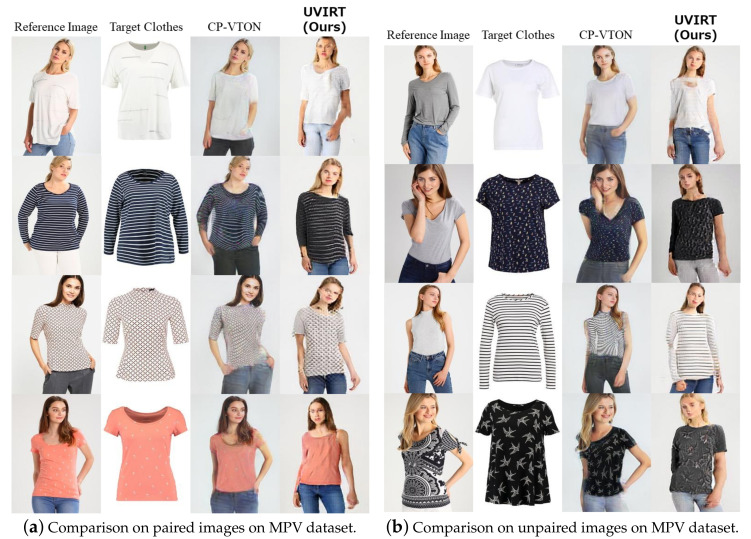
Qualitative comparison of proposed UVIRT vs. CP-VTON on the test images of the MPV dataset. The left figure shows the results of reconstructing the reference images using the target clothing. The right figure shows the results of warping target clothing to reference images. Although our UVIRT method changes the person’s poses, ours can warp clothing to the persons in each of the experiments. Thus, our UVIRT method achieves competitive performance in the virtual try-on task.

**Figure 4 sensors-20-05647-f004:**
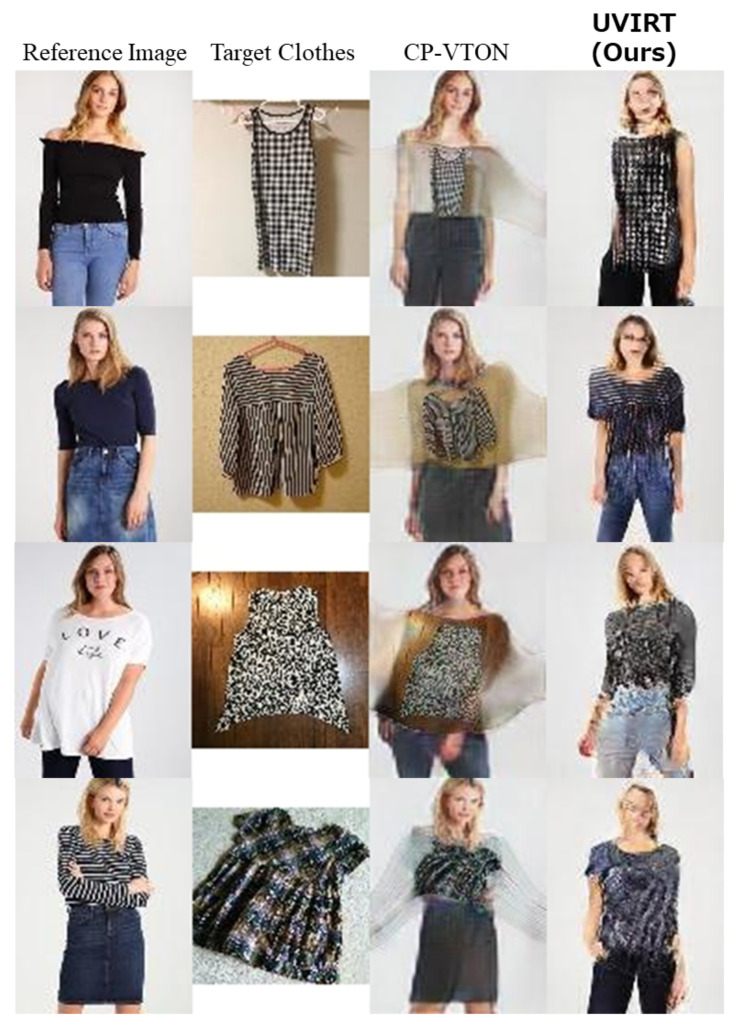
Qualitative comparison of Proposed UVIRT vs. CP-VTON on the test images of the C2C market dataset. CP-VTON cannot warp the unaligned clothing to the persons well. On the other hand, our UVIRT can warp the unaligned images to the person because of disentangling the person and the clothing features in the latent space.

**Figure 5 sensors-20-05647-f005:**
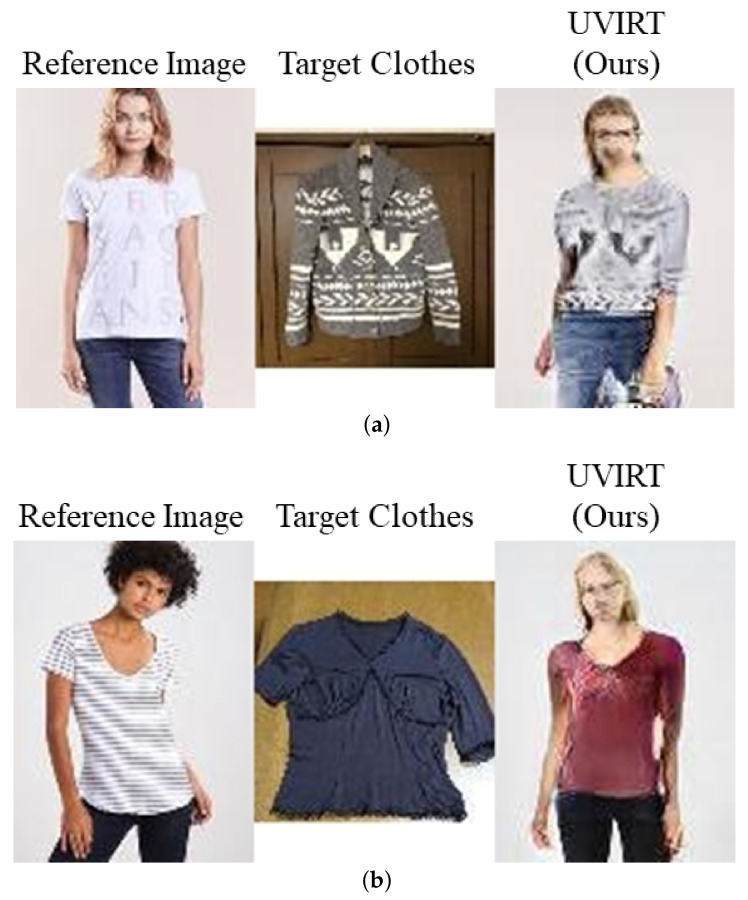
Failure case of our UVIRT. (**a**): The color of the target clothing change. Moreover, the face of the generated image is not reconstructed well because the generated image entangles the person and the clothing features. (**b**): Not only do the color of the clothing change but also the person changes because of the same reasoning discussed in (**a**).

**Table 1 sensors-20-05647-t001:** Quantitative comparison of proposed UVIRT vs. CP-VTON on the test images of the MPV dataset. Lower is better in the LPIPS and FID metrics, means and standard deviations are shown. The bold characters indicate the highest score.

Method	LPIPS ↓	FID ↓
CP-VTON (Supervised)	**0.191**	**25.23** (±0.17)
UVIRT (ours) (Unsupervised)	0.442	44.80(±0.21)

**Table 2 sensors-20-05647-t002:** Quantitative comparison of proposed UVIRT vs. CP-VTON on the test images of the C2C market dataset. Our UVIRT outperforms CP-VTON pre-trained with the MPV dataset.

Method	FID ↓
CP-VTON * (Supervised)	122.83 (±0.51)
UVIRT * (ours) (Unsupervised)	245.25 (±0.77)
UVIRT ${}^{†}$ (ours) (Unsupervised)	**92.55** (±**0.29**)

* methods were trained with the MPV dataset for fair comparison, namely the cross-dataset evaluation. ${}^{†}$ method was trained with the C2C market dataset.

**Table 3 sensors-20-05647-t003:** Comparison of the impact of the training data scale on the C2C market dataset. In the unsupervised setting, the C2C market dataset at 50% scale is superior to other scale settings.

Training Data Scale	FID ↓
5% (33,721 images)	95.01 (±0.26)
10% (67,443 images)	98.97 (±0.41)
25% (134,886 images)	99.31 (±0.60)
50% (269,773 images)	**92.55**(±**0.29**)
100% (539,546 images)	95.13 (±0.24)

**Table 4 sensors-20-05647-t004:** Testing time comparison of proposed UVIRT vs. CP-VTON on the test images of the C2C market dataset. We use an NVIDIA Tesla V100 in our tests.

Method	Testing Time [msec] ↓
CP-VTON (Supervised)	662.81
UVIRT (Unsupervised)	**53.94**

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
