# Peer review of "UVIRT—Unsupervised Virtual Try-on Using Disentangled Clothing and Person Features"

_sensors, 2020, doi:10.3390/s20195647_

Round 1
Reviewer 1 Report
Some minor editing changes suggested but overall the paper is very well presented. The concept of unsupervised learning for virtual try-ons is very interesting and this study gives a number of positive indications that more can be done to develop research in this area. The concept of applying disentanglement to this research area is also quite interesting and one which warrants further research.
Author Response
We really appreciate your review.
Please check our cover letter.

Reviewer 2 Report
In this paper, the authors propose a new unsupervised virtual try-on method using disentangled clothes and person features. The snsupervised approach is very useful because it allows to skip the labeling of different cloth types.
The manuscript is well written and structured overall. In addition, it is judged that an appropriate experiment has been performed even though a quantitative experiment is a research topic that can be difficult.
However, it is difficult to determine whether this paper fits the scope of the Journal - Sensors. Authors should show at least some of the papers that can resolve this issue.
Also, please make sure that there are no copyright issues about the clothing design in the figure of the paper.
Please review the English grammar throughout the paper.
As a result, the composition and contribution points of the manuscript are acceptable, but it is recommended to check the compatibility with the journal, copyright issues, and the use of English.
Author Response

(The authors gave the same response as above.)

Reviewer 3 Report
This manuscript proposes an unsupervised method based on disentanglement and convolutional autoencoders to provide a virtual clothes try-on. The method is very interesting and the idea of not needing annotated data is important. The newly collected database is also a good contribution. Nevertheless, the conclusion that the method’s performance is comparable with the state-of-the-art is not supported by the results of the somewhat faulty experiments. Also, some details need to be clarified and the manuscript needs some thorough proofreading.
- It is unclear what the method needs to be trained. It it necessary to have pairs of pictures with the same clothing piece (one with person and one without)? Doesn’t that require some type of supervision?
- This sentence in page 2: “(i.e.an only clothes image and a person images, who wear clothes)” is not understandable. The authors should clarify this.
- “However, the weakly-supervised virtual try-on method proposed by Pumarola et al. cannot be applied to model the clothes2person task. They can only apply their approach to transfer clothing between images that all contain people because the above two pre-trained networks only work on images containing people” The term clothes2person is not very clear. The authors should define such terms beforehand.
- The authors only use CP-VTON for comparison. This makes for a very weak evaluation section. Moreover, the proposed method, in Tables 1 and 2, only shows improved results over the alternative in the proposed database. This is a result of different evaluation settings: in Table 2, the CP-VTON is actually being evaluated in cross-database settings (much more challenging), while the proposed method is trained and evaluated in the same database. The authors should train UVIRT in MPV to obtain fair results.
- The proposed method seems to offer images with several distortions. This is visible in Figures 3 and 4, which should be enlarged for easier analysis by the readers. What do the authors think about these distortions? Will they affect real applicability in a virtual try-on product? What may be their cause? What do you plan on doing to address this problem.
- “We are the first to apply real-world C2C market data to virtual try-on. This is demonstrated through experiments on a real-world C2C market dataset.” Do the authors intend on providing a freely-accessible online demo of their method. Just a simple demo would be useful to check the potential of the method in first hand.
- The paper needs some proofreading. E.g., “is still not be applicable” and “cloth-parsing” in page 2.
Author Response

(The authors gave the same response as above.)

Reviewer 4 Report
A novel unsupervised virtual try-on method using disentangled representation is proposed.
Authors should highlight in the introductory section what are the performance advantages of the proposed method. While it is true that supervised approaches may require additional computational cost for annotation processes, they can nonetheless guarantee high quality results. Therefore, the authors must clarify in the introduction how the proposed unsuervised algorithm manages to guarantee, with lower computational costs, a high quality of the results.
The proposed method must be described in more detail in section 3. The architectural scheme in Fig. 1 requires a more detailed description than the one summarized in the legend of the figure. It is recommended to delete the process description in the legend of the figure, detailing it in section 3. Furthermore, in section 3 the algorithm that implements the proposed method must be added in structured pseudocode mode.
It is unclear what the number of images in each dataset is used for comparison testing. Is it large enough to represent a meaningful sample? Authors should make this point clear.
Finally, it is necessary that the authors also include comparative tests related to running time to show the advantages of the proposed method also in terms of computational time.
Author Response

(The authors gave the same response as above.)

Round 2
Reviewer 3 Report
I think the authors addressed most of my comments and doubts. However, I still believe Table 2 should include the evaluation of UVIRT trained only on MPV and tested on C2C, in order to have a fair comparison with CP-VTON in the exact same conditions. I understand that, since UVIRT is unsupervised, the authors can also keep the current results of train+test on C2C and mention it as advantages of the proposed method. I also consider paramount that the authors enlarge the images in Fig. 3 and 4.
Author Response
Thank you for your hard work on reviewing our paper.
We conducted the cross-dataset evaluation. Please check the Table 2. of our updated manuscript. CP-VTON was trained with the MPV dataset because it is a supervised method and can not be trained on the C2C market dataset. UVIRT was trained with the MPV dataset and the C2C market dataset for fair comparison. When both methods were trained on the MPV dataset, CP-VTON showed better performance on the C2C market dataset. However, when UVIRT was trained on the C2C market dataset (something that can not be done for CP-VTON), UVIRT demonstrated significantly better performance. The strong point of UVIRT is that it can be trained with in-the-wild datasets. Our UVIRT also achieves more high fidelity image generation than CP-VTON when trained with the C2C market dataset. Consequently, we have demonstrated our UVIRT is effective qualitatively and quantitatively in an in-the-wild dataset virtual try-on task.
Additionally, we enlarged the Fig. 3 and 4.
Reviewer 4 Report
The authors take into account all my suggestions, improving the quality of their manuscript. I consider this work publishable in the present form.
Author Response
We appreciate your hard work on reviewing our manuscript.